# Emitting long-distance spiral airborne sound using low-profile planar acoustic antenna

Shuxiang Gao [1], Yunbo Li [2], Chengrong Ma [1], Ying Cheng [1✉] & Xiaojun Liu [1✉]

Recent years have witnessed a rapidly growing interest in exploring the use of spiral sound carrying artificial orbital angular momentum (OAM), toward establishing a spiral-wave-based technology that is significantly more efficient in energy or information delivering than the ordinary plane wave technology. A major bottleneck of advancing this technology is the efficient excitation of far-field spiral waves in free space, which is a must in exploring the use of spiral waves for long-distance information transmission and particle manipulation. Here, we report a low-profile planar acoustic antenna to modulate wavefronts emitted from a near-field point source and achieve far-field spiral airborne sound carrying OAM. Using the holographic interferogram as a 2D modulated artificial acoustic impedance metasurface, we show the efficient conversion from the surface wave into the propagating spiral shape beam both numerically and experimentally. The vortex fields with spiral phases originate from the complex inter-modal interactions between cylindrical surface waves and a spatially-modulated impedance boundary condition. This antenna can open new routes to highly integrated spiral sound emitters that are critical for practical acoustic functional devices.

[1] Key Laboratory of Modern Acoustics, Department of Physics and Collaborative Innovation Center of Advanced Microstructures, Nanjing University, Nanjing, China. [2] State Key Laboratory of Millimeter Waves, Southeast University, Nanjing, China. ✉email: chengying@nju.edu.cn; liuxiaojun@nju.edu.cn

The introduction of artificial orbital angular momentum (OAM) has opened up new ways to control classic waves, which attract a rapidly growing interest in recent years toward establishing a spiral-wave-based technology that is significantly more energy or information efficient than the ordinary plane wave technology. Spiral wave field carrying OAM presents a helicoid phase dependence containing a screw-type phase singularity at the center, leading to the null pressure amplitude at the beam axis. At the same time, the helical phase wavefront can exert a measurable torque on the absorptive object[1–6]. These representative characteristics of the vortices can be useful for manipulating particles such as trapping[7–10], pulling[11–14], levitation[15,16], or data transmission[17–20] in both optics and acoustics. The inherent polarization of electromagnetic (optical) waves limits the efficiency of generated spiral field, making the spiral field more promising in acoustics. There have been substantial efforts to transpose the idea of helical wave from optics to acoustics. Among them, the pioneering design of synthesizing experimentally acoustical vortices with four synchronized transducers was first proposed by Marston et al.[21] and prominent efforts have been dedicated to developing relevant theories in this field[22,23] along with this seminal work. The conventional methods for generating acoustic OAM can be divided into two major categories: active and passive. Breakthroughs have been first made based on the active technology essentially using acoustic phased-array technology[24–26], resorting to a large number of individually addressed transducers with corresponding signal processing system, whereas the pioneering passive methods employ transmissive architecture, which is in need of either a bulky planar emitting transducer or large spatially-varying thickness[27,28] based on the acoustic diffraction effects.

On the other hand, the emergence of ultra-thin acoustic artificial structure, which is called metasurface, provides new ideas for implementing acoustic vortex. Compared with the traditional three-dimensional (3D) acoustic meta-material, two-dimensional (2D) metasurface has the advantages of low profile, small loss, and extreme control of phased-array wave field, and thus can effectively carry out the acoustic wavefront regulation[29–37] and microparticle manipulation[20,38]. Recently, a new mechanism of passive generated OAM is produced by employing the acoustic resonance, which is independent of thickness and effective propagating wavelengths[39]. It is expected that for applications such as fluid mixing, operation in the near field of this antenna can be useful. Although the requirements for miniaturization are met, the propagating distance in free-space airborne environment is still limited, about 2 wavelengths. To extend the sound energy to the far field, the sound wave needs to be transmitted in a waveguide[40], which is difficult to apply in practice. Otherwise, the entire system has to be embedded in fluid[41–43], suffering from the extreme impedance mismatch. Therefore, novel schemes are in great need to develop integrated functional devices for long-distance propagation of spiral sound in free space that can overcome the limitations of current acoustic technologies.

In this work, we report the discovery of an approach for the excitation of far-field spiral wave in free space by modulating the evanescent spoof surface acoustic wave (SAW). A low-profile planar acoustic antenna based on holographic impedance metasurface is constructed. As shown in Fig. 1, when the SAW field is radiated by an ordinary point-like sound source at the center, the structure can effectively regulate the phase delay of SAW to generate specific spatial wavefront pattern with the output phase at the exit end spirally distributed along the $\theta$ direction, whereby far-field acoustic spiral waves can be formed. It is noteworthy that synthesizing bulk acoustical vortices in near-field through Rayleigh SAWs generated by spiraling holograms was previously proposed in ref. [44], in which active holograms based on spiraling

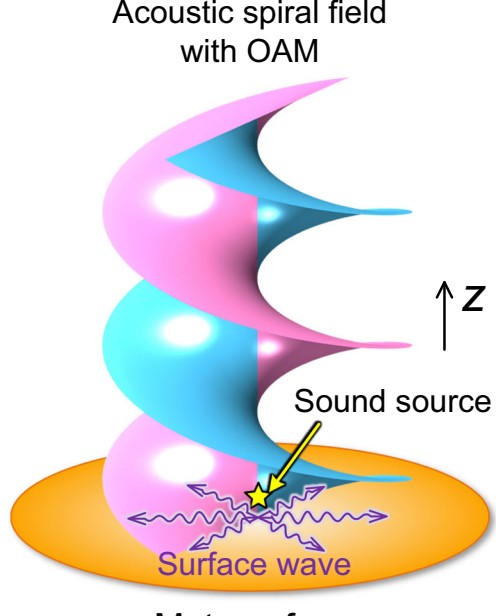

**Fig. 1 Illustration of the metasurface antennas.** The axisymmetric acoustic surface waves are launched by a monopolar sound source (yellow pentagram) located at the center of the metasurface antennas (orange plate), as illustrated by the purple wave packet representing the pressure field amplitude. These surface waves carrying zero OAM are scattered into the desired far-field wave beam with artificial OAM, depicted by the vortex phase fronts.

interdigitated transducer were used to encode the phase of the field like a hologram and shape SAWs instead of the metasurface proposed here.

## Results

**Theory of impedance metasurface.** The concept of SAW impedance is first put forward to play a key role as a bridge linking propagating waves and evanescent surface waves. According to the theory of holography[45], the phase information of the holography could be captured by interfering the scattered (radiation) wave from an object with a reference wave of the same frequency. For a reference wave $\psi_{ref}$ and an object wave $\psi_{obj}$, the holographic interferogram[46] is given by $\Psi = \left| \psi_{ref} + \psi_{obj} \right|$. In our approach, acoustic impedance is introduced to characterize the intensity gradient of the interferogram; thus, it can be rewritten as

$$Z = j[X + M\,\text{Re}\,(\psi_{ref}^* \psi_{obj})], \qquad (1)$$

where $X$ and $M$ are the average SAW impedance and modulating depth, respectively. When the interference pattern is excited by the reference wave, the desired object wave can be reproduced as $\psi_{ref}(\psi_{ref}^* \psi_{obj}) = \left| \psi_{ref} \right|^2 \psi_{obj}$.

To facilitate the implementation, the ideal continuous holographic impedance metasurface should be discretized into the discrete metasurface consisting of 2D modulated basic unit cells. Each cell as shown in Fig. 2a can be recognized as a rigid body with a subwavelength center-hollow-hole filled with background medium air[47–50], which is designed to couple the SAW modes with surrounding medium. By gradually changing the hole sizes, we can obtain the different surface refractive indexes and SAW impedances without breaking the quasi-periodicity of the whole structure. In this study, the eigen-mode analysis method in COMSOL Multiphysics is applied to extract the dispersive

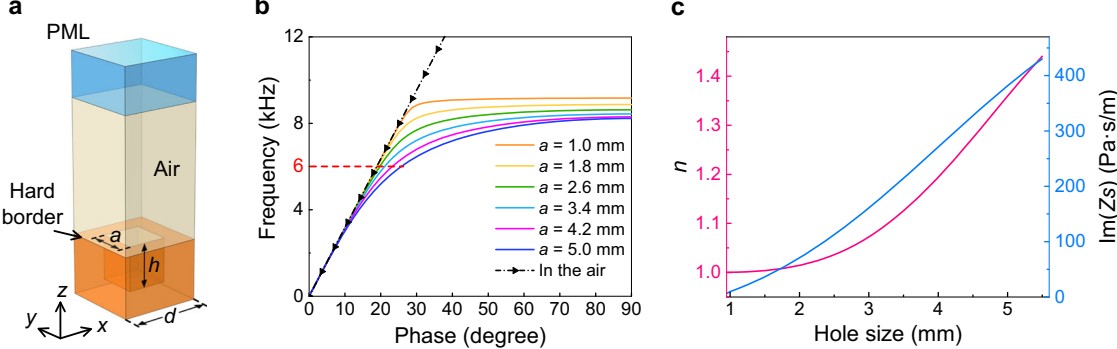

**Fig. 2 Building block used for artificial phase modulations. a** Basic unit cell: cuboid holes embedded in a solid base are arranged in square lattice, with the period of $d = 6$ mm, the hole depth of $h = 9$ mm, and the variable hole size $a$. **b** Dispersion curves of unit cells with different hole sizes $a$, while the black line with triangle marks denotes the air line. **c** Surface refractive index (pink) and surface impedance (blue) curves with changing hole sizes at 6000 Hz.

relation between the phase delay of a unit cell (with a fixed period of $d$) and the frequency (Fig. 2b), which is defined as

$$\phi = k_t d = k_0 n d, \qquad (2)$$

where $\phi$ corresponds to the phase difference across the unit cell, $k_t$ and $k_0$ are the wave numbers on the metasurface and in free space, respectively, and $n$ is the average surface refractive index of the metasurface. Based on Eq. (2), the effective surface refractive index can be directly acquired. Combining with the equation of SAW impedance (Supplementary Note 1), the relation between the phase offset and SAW impedance can be described as:

$$Z_s = Z_0 \sqrt{1 - n^2} = Z_0 \sqrt{1 - \left(\frac{\phi}{k_0 d}\right)^2}. \qquad (3)$$

Therefore, once the phase offset is acquired through the dispersion curve, the effective surface refractive index and SAW impedance can be calculated correspondingly (Fig. 2c). On the contrary, the hole size can be set using interpolation method, as specific hole size corresponds to a particular phase delay at a certain frequency, provided the phase delay is set by the determined impedance. For illustration, at the intended working frequency of 6000 Hz, the height of the hole is 9 mm, which can be regarded as subwavelength, and the hole size varies from 2 to 4 mm with the values of $X$ and $M$ determined as 170.9 and 99.8, respectively.

To complete the design of a 2D modulated artificial acoustic metasurface, the SAW impedance textures of the surface should be first shaped. Assuming that a cylindrical sound source is located at the origin on the metasurface, which occupies the $X$–$Y$ plane, the reference wave (or the surface plane wave) can be written as

$$\psi_{\text{ref}} = e^{-jk_0 n r}, \qquad (4)$$

where $r$ is the radial distance from the center origin. The object wave is defined as[51]

$$\psi_{\text{obj}} = e^{-jk_0 r \sin \theta_0}, \qquad (5)$$

where $\theta_0$ is the desired radiation elevation angle. Hence, the hologram composed of impedance can be expressed as

$$\begin{aligned} Z_s &= j[X + M \operatorname{Re}(\psi_{\text{ref}}^* \psi_{\text{obj}})] \\ &= j[X + M \cos(k_0 n r - k_0 r \sin \theta_0)]. \end{aligned} \qquad (6)$$

To obtain the periodicity of the holographic distribution of surface impedance along one direction, we make the phase item satisfy $(k_0 n - k_0 \sin \theta_0)p = 2\pi m_0$, so the corresponding period is

represented by

$$p = \frac{2\pi m_0}{k_0 n - k_0 \sin \theta_0}, \qquad (7)$$

which can be rewritten as

$$k_0 \sin \theta_0 = k_0 n - \frac{2\pi m_0}{p}. \qquad (8)$$

Here the left side of Eq. (8) represents the radiation term, whereas the first and second term of the right side are the wave vector of SAW and the modulation term, respectively. In essence, the shift of wave vector transfers the surface wave to the radiation wave (Supplementary Note 2). It is noteworthy that what we need is the $-1$ order diffraction term only, that is to say, when $m_0 = 1$, the left side of the equation should be modulated into the radiation area (i.e., point I to II); but when $m_0 = 2$, this term need to be removed out of the air cone (i.e., point I to III).

**Acoustic spiral field**. In a spiral field, the wave will twist along its axis as it travels, forming a beam shape similar to a bottle screw. The propagation phase satisfies:

$$\phi_{\text{obj}}(\theta) = e^{-jm\theta}, \qquad (9)$$

in which the expression about the exit direction is set to be 0 ($\sin 0°$) as the desired direction of radiation is along the normal of the metasurface, whereas $m\theta$ acts as an additional phase depending on the azimuth angle, to generate a twisted wavefront. On the basis of the previous derivation, the SAW impedance interferogram here is given by

$$Z_s = j[X + M \cos(k_0 n r - m\theta)], \qquad (10)$$

where $m$ is called the topological charge or the order of the spiral field, which is defined as the number of twist of the wavefront within one wavelength distance. For generating a first-order vortex beam with $m = 1$, the holographic pattern is shown in Fig. 3a, which is combined with the dispersion and impedance data shown in Fig. 2b, c, to determine the required hole as a function of spatial position. The pattern is a spiral-shaped strip and the spiral direction of the outgoing beam is consistent with the rotation of the surface spiral impedance strip.

We construct a sample antenna of finite dimension, whose profile of hole size $a$ in the circumference and radial directions along pink dashed circle and blue dashed hatching line labeled in Fig. 3a are shown in Fig. 3b, c, respectively. The whole sample is a round sheet with a radius of 30 cm, composed of 7660 unit cells. Smaller transverse dimensions and the limitations are discussed in Supplementary Note 3. Measurements are designed and carried out to validate the generation of far-field spiral waves by the

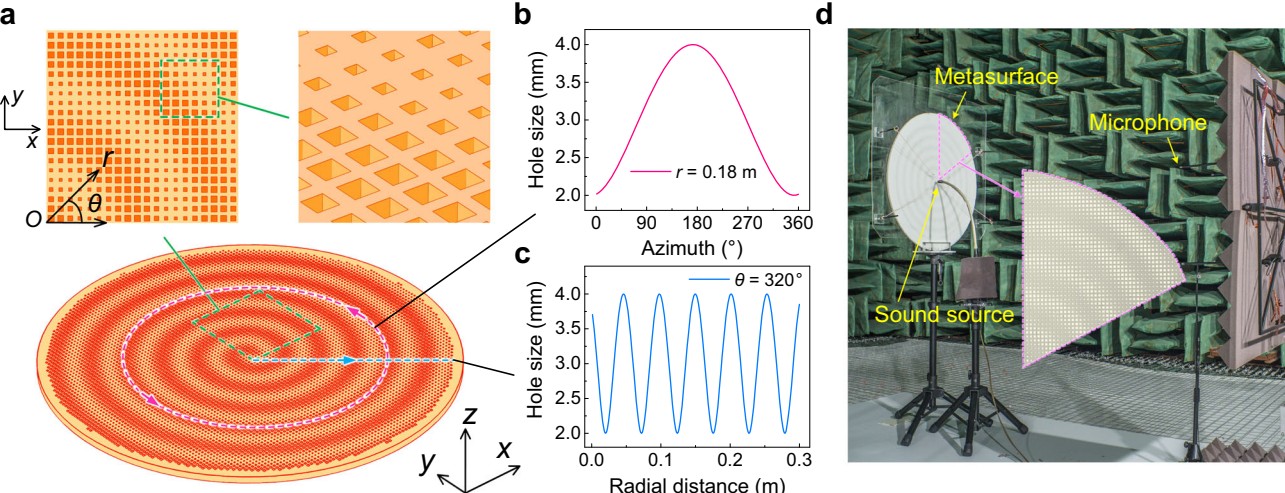

**Fig. 3 Planar acoustic antenna design. a** Schematic diagram. The structure details in the green dashed squares are shown in expanded schematic form (square insets). **b**, **c** Plot of the geometric parameters that are implemented. The pink solid (blue solid) curve illustrates the profile of hole size $a$ along the pink dashed circle (blue dashed hatching line) in the circumference (radial) direction. **d** Experimental setup. The fanshaped inset shows partial enlarged view.

proposed method in the anechoic chamber and the photograph of experimental setup is illustrated in Fig. 3d. In fabrication, the sample material is chosen to be photosensitive resin (via the 3D printing technology), which can be treated as acoustically rigid for airborne sound. In measurements, we employed a point-like source placed at a quarter wavelength away from the metasurface to excite the SAW, whose directivity is similar to that of an electromagnetic dipole in accordance with mirror image principle. The far-field radiation information is detected by microphone array (G.R.A.S. Type 40PH), moving step by step in the plane parallel to the metasurface, with one additional microphone fixed in the outgoing field as a reference signal. As a result, the amplitude and phase of the sound in each scan point can be recorded in real time.

The corresponding simulation results are shown in Fig. 4 and the right-handed radiation wave can be received in the $z$ direction. The simulation is carried out in a cylindrical calculation domain with the designed metasurface positioned at the bottom of the domain and the spatial lateral extent of the simulation window is selected to be 0.34 m, which is about $0.7\lambda$ beyond the radius of the metasurface. The cylindrical wave radiation condition is applied on the exteriors of the domain to prevent boundary diffraction at the edges, whereas a perfect matching layer is employed to absorb the radiated sound waves in the far field. Figure 4a illustrates a cutaway view of the outgoing sound field with the same radius of the metasurface. Good collimation characteristics can be observed along the entire propagation axis, making it possible to form a stable acoustic vortex beam over a considerably long distance. For illustration, we only depicted the spiral sound field within 2.00 m, which has reached 35 spatial wavelengths already. Given that the near field is inevitably subject to the interference of surface wave, the critical distance between the near- and far-field is defined as $z = R^2/\lambda = 1.57$ m referring to the radiation of a circular piston[52], and the far field exhibits the perfect acoustic vortex. The sound energy of the vortex is obtained by integrating the intensity $I_z$ of the horizontal section of the cylindrical calculation domain in the far-field and its ratio to the total energy radiated from sound source obtained by integrating the sound intensity over the entire boundary of the model determines the efficiency of energy conversion as 76%. In

Fig. 4b, a section of the far-field region is enlarged to manifest the distortion of both amplitude and phase more clearly (see Supplementary Movie 1 for the dynamic view of acoustic vortex beam). From the cross-sectional view, the phases have shifted from $-\pi$ to $\pi$ in each turn, revealing the expected topological charge $m = 1$. Moreover, the superposition of the phases at the center results in an intensity minimum at $z$ axis (point B), which can be also seen in Fig. 4c. By contrast, sound energy is well concentrated in other areas of circular wavefront (i.e., point A) so that the beam can reach the far field, even though for each wavelength the wavefront of the vortex propagates forward, the sound pressure amplitude will decrease by around 4% as a result of the inevitable viscous loss in experiments. In addition, the measured phase distributions and sound amplitudes at the far-field cross-sections $z = 1.57$ and 2.00 m are shown in Fig. 4d (see Supplementary Note 4 for measured evolution process of phase diagrams within one period). The expected OAM in a first-order spiral beam with a smooth helical phase and null amplitude at the core can be clearly observed. Measurement results have good agreements with the numerical simulations, which demonstrates the performance of metasurface antenna in converting acoustic surface wave to a radiation beam with OAM. The weak asymmetry in the amplitude originates from the discretization differences among the impedances in each direction, which leads to the spiral stripes in impedance, despite the superficial resemblances of cylindrical symmetry. Consequently, the impedances traversed by the surface waves on each path are not identical to bring the corresponding outgoing wave field with an azimuthal angular dependence of $e^{-jm\theta}$, which introduces some unevenness in the output amplitude.

From the above discussions, we can see that the proposed method achieves the desired first-order vortex beam both theoretically and experimentally. This method can be further extended to higher-order vortex beams by directly assigning a larger value to $m$. Figure 5 presents the amplitude and phase distributions of wavefront of the far-field vortex beam with topological charges $m = 2$–4 (see Supplementary Movie 2 for corresponding dynamic view). As in the previous section, the cross-sections at 2 m confirm that the long-distance propagation of the spiral beam could be maintained via adding the number of

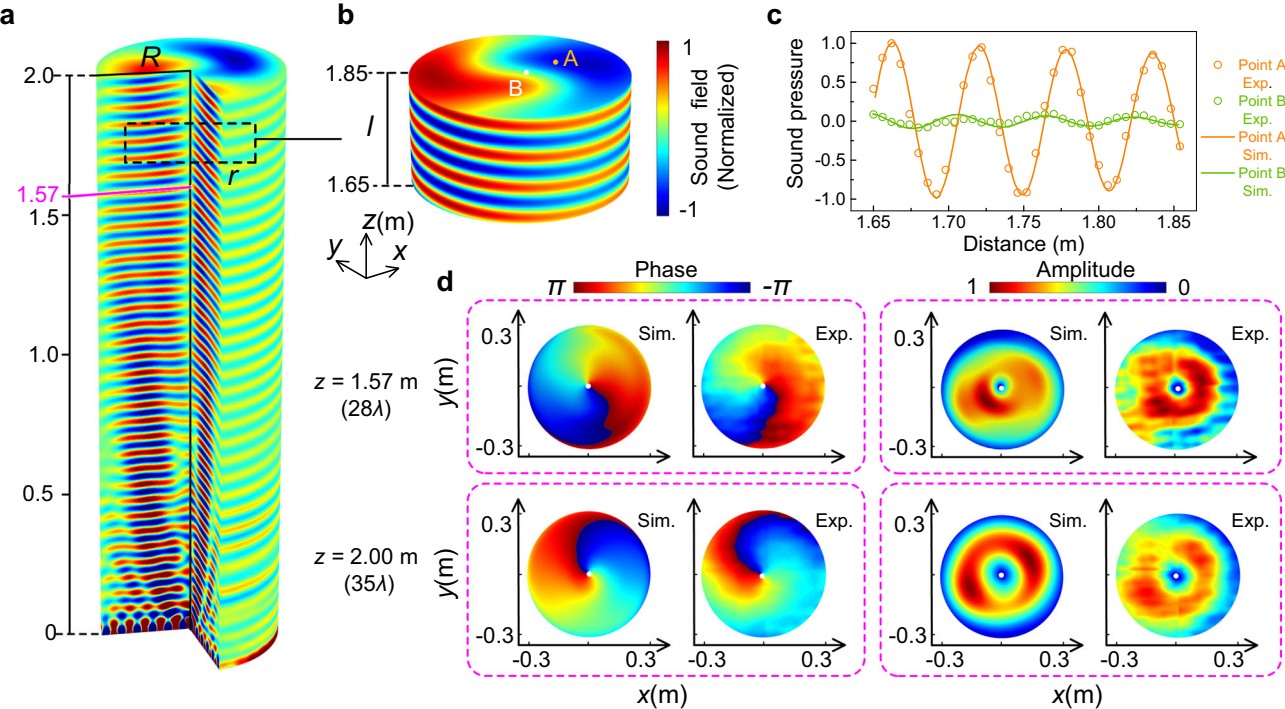

**Fig. 4 Far-field spiral sound in free space. a** Normalized sound pressure field along the propagation axis in cutaway view. The cutting position is the cylinder with the radius $R = 0.3$ m. The transition point between the near and far field is around 1.57 m. **b** Zoom-in of the far-field rotating wavefront, which is taken from a region with radius $r = 0.2$ m and $l = 0.2$ m, located at $z_0 = 1.65$ m (black dashed line in **a**). **c** Experimentally measured (Exp.) and simulated (Sim.) sound pressure profiles along the vertical lines across point A and B in **b**. **d** Experimentally measured and simulated phase and amplitude distribution of the helical wavefront at two observation planes located at 1.57 and 2.00 m away from the metasurface, respectively. The white dots refer to the geometric centers of the phase cross-sections.

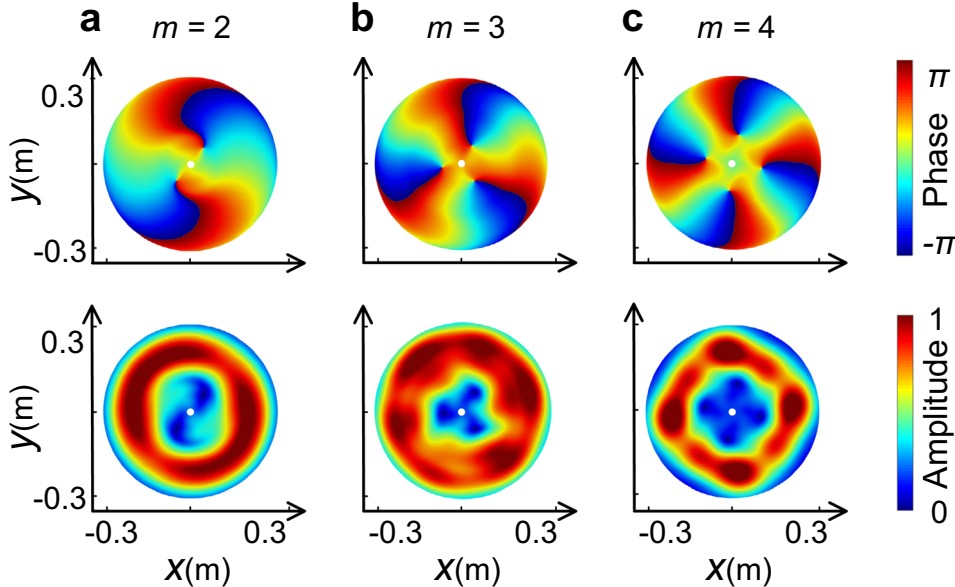

**Fig. 5 High-order far-field vortex beams. a–c** Show the vortex beams with OAM topological charges $m = 2$–$4$. Top and bottom row show the phase and amplitude distribution of the pressure field, respectively.

turns. It is noteworthy that we concentrate on the modulation of the output phase in this work. Although the radial distribution of the amplitude in theoretical design is set to be uniform, the actual amplitude values on each path are not exactly identical due to the impedance modulation and the quite modest difference can be evaluated from the stable vortex beams in far-fields in simulations and experiments. In addition, in the case of $m > 1$, the phase

rotation with the azimuth angle is increased, where the helical wavefront reaches a total phase shift of $2\pi m$ over a complete turn. Hence, the phase singularity still emerges at the core without offset, while the hollow part of the central area is extended arising from the variation of the phase, indicating that the vortex of charge $-m$ has been split into $m$ vortices of charge $-1$ rather than 1. This phenomenon can be ascribed to effect that the high-

order vortex has a more unstable structure than the first-order vortex and some slight disturbances could degenerate the unstable high-order vortex into multiple stable first-order vortices[53,54]. In our simulations, the disturbances mainly arise from the weak non-cylindrically symmetric structure design and the interference of the point-like sound source with the radiated acoustic vortex[54,55].

## Discussion

Derived from its extraordinary properties of effective surface impedance, the planarized vortex antenna exhibits high efficiency in the modulation and transformation of evanescent waves into complex far-field radiated sound waves, which may promise extensive and significant application advantages. Although we mainly focused on the generation of long-distance spiral beam in this work, the proposed strategy also offers a feasible approach to design an integrated and small-sized system-level antenna for realizing versatile functions with superior performance far beyond the scope of this work, as well as being loaded on other conformal functional devices of irregular shapes. For example, the ability of far-field collimated vortex propagation with stable topological charge can be used in non-contact manipulation of particles and acoustic long-distance communication. Moreover, our design scheme of holographic impedance metasurface can also be extended to acoustic multi-mode communications as a consequence of the flexible construction of surface impedances and integration of different modes such as various angular momentums, working frequencies, and outgoing beam angles, which have promising application prospects in the field of signal multiplexing (e.g., see Supplementary Figs. 6 and 7 for illustration of dual-frequency and dual-angle OAM-dependent acoustic antenna, respectively; detailed discussion can be found in Supplementary Note 5).

In summary, we report an approach to building a holographic impedance metasurface that can associate acoustic surface waves with spatial waves and realize far-field radiation beams carrying OAM. The underlying physics of wavefront control and the stable topological charge is demonstrated in a considerably long distance in the propagation direction, through both simulations and experiments. With the compact profile, long propagation distance, controllable stable topological charge, and easy-to-manufacture, the intriguing metasurface antenna stands in contrast to previous transmissive designs, which may have many potential applications in future integrated functional devices.

## Data availability

The data that support the findings of this study are available from the corresponding authors on reasonable request.

## Code availability

The code used to calculate the results for this work is available from the corresponding author upon reasonable request.

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

## Acknowledgements

This work was supported by the National Basic Research Program of China (2017YFA0303702) and NSFC (12074183, 11922407, 11834008, and 11874215).

## Author contributions

Y.C. initiated the project. Y.C. and X.L. guided the research. S.G. and Y.L. conceived the design. S.G. and C.M. carried out the theoretical modeling and FEM simulations, designed the experimental setup, and conducted the measurements. All authors contributed to the data analysis and manuscript writing. S.G. and Y.L. contributed to this work equally.

## Competing interests

The authors declare no competing interests.
