## [Peer Review File · Nature Communications]

Reviewers' comments:

Reviewer #1 (Remarks to the Author):

In this paper, the authors propose a method based on 2D reflective metasurfaces to shape surface acoustic waves and generate helical waves in air. The paper is interesting and technically sound. The acoustic fields synthesized with this technique are properly characterized with an array of microphones and experiments fit reasonably well with numerical predictions. The paper is clear, concise and comprehensive and gives all the required information to reproduce the results. Thus, from a technical point of view, this paper deserves publication, providing that the authors answer to the questions below.

Nevertheless, the proposed method remains one method among a wealth of other existing methods to synthesize acoustical vortices. Its superiority or particular interest for some specific applications is not demonstrated. For example, Marzo and coworkers (ref [18] of the paper) showed that acoustical vortices can be synthesized in air with a relatively cheap array of transducers and electronics, which, in addition, is more versatile. So, what are the new capabilities offered by metascreens? In particular, there is no data provided on the energy conversion from the source to the vortex, while the abstract mentions "an efficient energy conversion from the surface wave into the propagating spiral shape beam".

So, I have mixed feelings about this paper. While the idea is interesting, the relevance and importance of metascreens for applications is not clear.

Concerning the technical content of the paper, I have a few questions:

- When the target field is defined, only its phase is mentioned. There are some stable forms of helical waves over propagation like Bessel beams, but they require a specific evolution of the amplitude as a function of the lateral radius (Bessel functions). So, is the lateral evolution of the amplitude controlled? If not, how stable the beam is expected to be?
- When we look at Fig. 4d, there is a certain anisotropy in the amplitude of the vortex as a function of the angle θ (even in the simulations), while the metasurface seems symmetric. What is the reason for this anisotropy?
- In the section "Simulations and experiments", the method to measure the acoustic field is exposed but there is no information on the simulations. The main ingredients of the method used for the simulations should be briefly exposed in the main text. In particular, what is the spatial lateral extent of the simulation window, and how is the diffraction at the edges (since the source is finite) handled?

Also, an important point is that the bibliography does not give credits to some relevant work of the community:

First, one of the most prominent researchers of the acoustical vortices community is Pr. Marston. He was the first to transpose the idea of helical wave from optics to acoustics. He was also the first to synthesize experimentally acoustical vortices with four

synchronized transducers in ref. [1*]. Along with this seminal work, he dedicated a large effort to develop relevant theories in this field, see. e.g. ref. [2*,3*] and I believe that some of his papers are relevant here.

Then, the first authors who synthesized an acoustical vortex with an array of transducers were Thomas and Marchiano in ref. [4*] in 2003, long before ref. [20,21] cited in the paper. So, this work should be mentioned along with the other two references.

In the present paper, the authors propose a method to shape SAW with metascreens in order to generate acoustical vortices. While this idea is undoubtedly original, the idea to use SAWs generated by spiraling holograms to synthesize bulk acoustical vortices was previously proposed in ref. [5*]. In this case, active holograms based on spiraling electrodes were used to shape SAWs instead of the metascreens proposed here. This work should be introduced properly.

Finally, concerning particles trapping and manipulation, ref [6*] demonstrated particles manipulation with helical waves with unprecedented selectivity. I suggest to introduce it along with ref. [7-9] of the paper.

[1*] B. T. Hefner and P. L. Marston, An acoustical helicoidal wave transducer with applications for the alignment of ultrasonic and underwater systems, *J. Acoust. Soc. Am.* 106, 3313 (1999).

[2*] P.L. Marston. Axial radiation force of a Bessel beam on a sphere and direction reversal of the force, *J. Acoust. Soc. Am.*, 120, 3518 (2006)

[3*] P.L. Marston Scattering of a Bessel beam by a sphere: II helicoidal case and spherical shell example, *J. Acoust. Soc. Am.* 124:5,2905–10 Marston (2008).

[4*] J.-L. Thomas and R. Marchiano, Pseudoangular Momentum and Topological Charge Conservation for Nonlinear Acoustical Vortices, *Phys. Rev. Lett.* 91, 244302 (2003).

[5*] A. Riaud, M. Baudoin, O. Bou Matar, L. Becera, J.-L. Thomas, Selective manipulation of microscopic particles with precursor swirling Rayleigh waves, *Phys. Rev. Appl.*, 7: 024007 (2017)

[6*] M. Baudoin, J.-C. Gerbedoen, A. Riaud, O. Bou Matar, N. Smagin, J.-L. Thomas, Folding a focalized acoustical vortex on a flat holographic transducer: miniaturized selective acoustical tweezer, *Science Adv.*, 5: eaav1967 (2019)

Reviewer #2 (Remarks to the Author):

Sound vortices, which carry quantized topological charges in real space, are very useful in real applications, such as for robust acoustic communication and rotating small particles. This paper provides a novel design route to achieve spiral sound field with long propagation distance. The approach is straightforward and efficient. The theoretical results are well supported by experiments. The paper deserves to be published in *Nature Communications*. Suggestions and comments are listed below.

1. The discussion part should be more concise. Some recurrent sentences should be removed.
2. I am puzzled about the sentence "Hence the phase singularity still emerges at the core without offset while the hollow part of the central area is extended arising from the variation of the phase." It is very nice that, in all cases, the phase winding number along a relatively large loop surrounding the weak field region emerges as desired. It seems that each vortex of charge- m is split into m vortices of charge-1, due to the presence of various experimental noise and errors. You can check this effect through a simulation with some fictitious noise, and provide a comment on this issue.
3. As a whole, the paper is well organized. However, the language should be polished further. In addition, some terms look strange. For example, "long transmission distance" should be "long propagation distance". It is better to use "hole depth" and "hole width", rather than "Air gap depth" and "air gap size", if the holes are defined to be square.

Reviewer #3 (Remarks to the Author):

This paper discusses an impedance metasurface formed by cuboid air gap for the generation of acoustic orbital angular momentum (OAM). While the authors claim that they can efficiently excite the OAM field, there is no significant advancement in physics or application. With the existing literature for the efficient generation of OAM both actively using transducer arrays [Hefner and Marston, J. Acoust. Soc. Am. 106, 3313-3316 (1999), Riaud et al, Phys. Rev. Appl. 4, 034004 (2015)] and passively with metasurfaces [Jiang et al, Phys. Rev. Lett. 117, 034301 (2016), Jiang et al, Appl. Phys. Lett. 108, 203501 (2016), Ye et al, AIP Adv. 6, 085007 (2016), Naify et al, Appl. Phys. Lett. 108, 223503 (2016)], I cannot find any improvement compared with these methods. This article does not meet the standard of a high quality journal like Nature Communications. Therefore, I cannot recommend it for publication.

Response to the Reviewers

We appreciate the reviewers for carefully reading our manuscript. The reviewers' comments are highly insightful and enable us to greatly improve the quality of our manuscript. We are strongly encouraged by reviewer #1 and reviewer #2's reports that the paper deserve publication for Nature Communications. We agree with the reviewers' comments and have revised the manuscript accordingly. We also thank reviewer #3 for the constructive comments and have gone through the entire paper for improvement, including additional results and discussions to address the comments completely. Here we include our responses to the reviewers' comments and denote the changes made in the manuscript. For the sake of readability, the changes to the manuscript are printed in red font.

We sincerely apologize for the unexpected delay due to the influence of COVID-19.

Response to Reviewer 1

Comment P 1.1 — *In this paper, the authors propose a method based on 2D reflective metasurfaces to shape surface acoustic waves and generate helical waves in air. The paper is interesting and technically sound. The acoustic fields synthesized with this technique are properly characterized with an array of microphones and experiments fit reasonably well with numerical predictions. The paper is clear, concise and comprehensive and gives all the required information to reproduce the results. Thus, from a technical point of view, this paper deserve publication, providing that the authors answer to the questions below.*

Response: We respectfully thank the reviewer for the valuable and constructive comments, which have helped us a lot in improving our manuscript clarity. We are strongly encouraged by reviewer's report that the paper deserve publication for Nature Communications. We are delighted to address the items raised by the reviewer.

Comment P 1.2 — *Nevertheless, the proposed method remains one method among a wealth of other existing methods to synthesize acoustical vortices. Its superiority or particular interest for some specific applications is not demonstrated. For example, Marzo and coworkers (ref [18] of the paper) showed that acoustical vortices can be synthesized in air with a relatively cheap array of transducers and electronics, which, in addition, is more versatile. So, what are the new capabilities offered by metascreens? In particular, there is no data provided on the energy conversion from the source to the vortex, while the abstract mentions "an efficient energy conversion from the surface wave into the propagating spiral shape beam".*

Response: We thank the reviewer for raising this important point and giving us the opportunity to clarify the problem. Indeed, we agree with the reviewer that the literature mentioned (Ref. [18] of the paper) does demonstrate a cheap array of transducers and electronics, which provides promising method for versatile applications such as manipulation of levitated objects. We have added contents to address this point. Here, we would like to note that our proposal is quite different from this reported active method, in terms of both physical mechanism and potential application scenarios.

Active method is essentially an acoustic phased array technique that uses multiple independently regulated acoustic transducers to form the array, producing reconfigurable spatial phase distribution of the desired spiral shape. For example, Ref. [18] demonstrated a 20×20 flat array, with each transducer 10 mm in diameter (that is, the element size is larger than the operating wavelength of about 8.5 mm at 40 kHz). However, in order to pursue a nearly continuous spiral wavefront for long-distance self-collimated propagation, it is often necessary to depend on a huge number of transducers forming an extremely large-scale acoustic array, and to operate individual units independently through complicated electric system composed of stimulant signal generation and power amplification, ensuring that the phase of the drive signal corresponding to each transducer is precisely modulated and powerful enough. Therefore, the active generation method of acoustic OAM requires relatively large space occupation and complicated circuit, and each transducer unit itself has a certain size, which also brings difficulties to its application in the high-frequency range.

On the other hand, the metasurface we proposed relies on passive technique that overcomes the inherent limitations of complex circuit control associated with the structure, but instead creates a spiral phase distribution required for the vortex field by using a metasurface with a discontinuous impedance gradient, which has the advantages of flat structure and relatively lower cost. Moreover, the fabricated metasurface in the manuscript is composed of 7660 elements, with the deep-subwavelength element period of $d = 6$ mm (that is, about $\lambda/10$, far smaller than the operating wavelength of about 56.7 mm at 6 kHz).

Furthermore, please note that the proposed method is also significantly superior in passive methods. Firstly, our metasurface does not depend on resonances produced by elements with composite internal micro-structure. Secondly, the acoustic diffraction effects require that the dimensions of the structure in the thickness direction should be greater than or equivalent to the acoustic wavelength, while our structure is an unconstrained flat, ultra-thin structure. Although all current planar structures claimed to be thin themselves, we would like to note that the excitation sound sources of incident plane waves and the required propagation distance to avoid near-field effects of practical transducers inevitably take up bulky space in either reflective or transmissive configurations, as these designs are based on the modulation of spatial propagating waves. What we are focusing on is the evanescent spoof surface acoustic wave, which is a guided wave pattern that only propagates along the surface of the rigid periodic structure on the fluid side. Therefore, our point excitation source can be positioned in the close vicinity of the metasurface, which functions as a coplanar structure as a whole, realizing a **real low profile**.

Energy conversion: we agree with the reviewer that the energy conversion should be investigated. To further corroborate the remarks, we have tested the total energy radiated from sound source by integrating the sound intensity over the entire boundary of the model. The sound energy of the vortex in the far-field is obtained by integrating the intensity I_z of the horizontal section of the cylindrical calculation domain, and its ratio to the total source energy determines the efficiency of energy conversion as 76 %.

Following the reviewer's suggestion, we have now revised the manuscript and made the above points clear to readers: "The sound energy of the vortex is obtained by integrating the intensity I_z of the horizontal section of the cylindrical calculation domain in the far-field, and its ratio to the total energy radiated from sound source obtained by integrating the sound intensity over the entire boundary of the model determines the efficiency of energy conversion as 76 %" (Page 5), "Although we mainly

focused on ... detailed discussion can be found in *Supplementary Information Note 5*)” (Page 6), and the detailed contents have been added in **Supplementary Information Note 5**.

Comment P 1.3 — *So, I have mixed feelings about this paper. While the idea is interesting, the relevance and importance of metascreens for applications is not clear.*

Response: We thank the reviewer for raising this important issue about the application potential of the proposed metasurface. Due to its unique air-compatible conductivity, ultra-thin structure and ability to carry sub-wavelength information to far fields, this metasurface may lead to a wide range of extensive and significant applications. Different from other OAM generators, the air-compatible conductivity allows the sound wave to propagate to a remote distance in the far-field without resorting to the additional waveguides (please note that these waveguides are ordinarily employed in most current metasurface designs for generation and transmission of acoustic vortex, but not applicable or cumbersome in many practical scenarios), which can be used in non-contact manipulation of particles, long-distance acoustic communication and non-intrusive detection of the energy flux, and the evolution of acoustic waves in the complex fluctuating system.

In addition, the ultra-thin structure makes it possible to produce compactor antenna because the required space for the source coplanar with the structure is minimized, which can only be accessed by using the spoof surface acoustic waves as the incident wave. The ultra-thin structure will provide a feasible approach to design integrated and small-sized system-level antenna as well as being loaded on other functional devices of irregular shape. Moreover, another important potential application is for far-field super-resolution imaging. The traditional OAM generators cannot carry sub-wavelength information, which will cause information missing in acoustic imaging. However, with this wave vector shifting device, the sub-wavelength information could be carried in the surface wave and then transferred to far fields. This offers the possibility to make far-field imaging without missing any information. The last but not the least, our design method of holographic impedance metasurface can also be extended to acoustic multi-mode communication as a consequence of the flexible design of impedance surfaces and integration of different modes such as different angular momentums, which has broad application prospects in the field of signal multiplexing.

Following the reviewer’s suggestion, we have also added brief explanations about above-mentioned relevance and importance of metasurface for applications on **Pages 6**: “*Although we mainly focused on ... detailed discussion can be found in *Supplementary Information Note 5*)*”, and the detailed contents have been added in **Supplementary Information Note 5**.

Comment P 1.4 — *Concerning the technical content of the paper, I have a few questions:*

- *When the target field is defined, only its phase is mentioned. There are some stable forms of helical waves over propagation like Bessel beams, but they require a specific evolution of the amplitude as a function of the lateral radius (Bessel functions). So, is the lateral evolution of the amplitude controlled? If not, how stable the beam is expected to be?*

Response: The reviewer brought up a very valuable point on the control of the amplitude of the vortex. We agree with the reviewer that enabling stable helical waves like Bessel beams usually require

a specific lateral evolution of the amplitude. In addition, it was reported that the Bessel-like helical waves could also be generated by using phase-only holograms without requiring a specific evolution of the lateral amplitude [see Sci. Rep. 9, 20104 (2019)], and the generated Bessel beams were stable with an uniform field distribution along the propagation distance. In our work, the theoretical design of the amplitude along the radial distribution is set to be uniform, but due to the impedance modulation, the amplitude values on each path are not exactly identical. However, the difference is quite modest, which can be evaluated from the good results of vortex beams in our simulation and experiment. Generally speaking, this work only concentrates on the design of the output phase and the amplitude does not need to be controlled.

Based on the reviewer's suggestion, above contents have been added in the revised manuscript: "Note that we concentrate on the modulation of the output phase in this work. Although the radial distribution of the amplitude in theoretical design is set to be uniform, the actual amplitude values on each path are not exactly identical due to the impedance modulation, and the quite modest difference can be evaluated from the stable vortex beams in far-fields in simulations and experiments" (Page 5).

Comment P 1.5 — - *When we look at Fig. 4d, there is a certain anisotropy in the amplitude of the vortex as a function of the angle theta (even in the simulations), while the metasurface seems symmetric. What is the reason for this anisotropy?*

Response: The reviewer has raised a good point that needs to be clarified to avoid confusion. The metasurface just looks cylindrically symmetric but has spiral stripes on it. As stated in above comment, the impedance metasurfaces traversed by the surface waves on each path are not the same in order to bring the corresponding outgoing wave-field with an azimuthal angular dependence of $e^{-jm\theta}$. This difference leads to some unevenness in the output amplitude.

Following the reviewer's suggestion, the contents to emphasize this issue have been added to the revised manuscript (Page 5): "The weak asymmetry in the amplitude originates from the discretization differences among the impedances in each direction, which leads to the spiral stripes in impedance despite the superficial resemblances of cylindrical symmetry. Consequently, the impedances traversed by the surface waves on each path are not identical in order to bring the corresponding outgoing wave-field with an azimuthal angular dependence of $e^{-jm\theta}$, which introduce some unevenness in the output amplitude".

Comment P 1.6 — - *In the section "Simulations and experiments", the method to measure the acoustic field is exposed but there is no information on the simulations. The main ingredients of the method used for the simulations should be briefly exposed in the main text. In particular, what is the spatial lateral extent of the simulation window, and how is the diffraction at the edges (since the source is finite) handled?*

Response: We thank the reviewer for pointing out this important issue. We totally agree with the reviewer that there should be a description of the simulation method.

The simulation area is a cylinder with the designed metasurface positioned at the bottom of the domain. The radius of metasurface is 0.3 m (50 d), and the spatial radius of the simulation window

(the radius of cylindrical calculation domain) is selected to be 0.34 m, which is about 7λ beyond the radius of the structure. The cylindrical wave radiation condition is applied on the exterior boundaries of the cylinder to prevent reflection. After multiple simulations, it is found that if the window radius exceeds the outer radius of the structure by more than 3λ , the simulation results are hardly affected by the boundary diffraction. This is in line with our expectations. Owing to the good spatial collimation of the structure, almost all of the converted radiation waves are parallel to the z-axis, ensuring that sound energy is concentrated in the circular plane of the structure size in the lateral direction. The selected simulation result shown in the manuscript is extracted from a cylindrical domain with a bottom radius of 0.3 m intercepted from the large cylinder. In the far field, a perfect matching layer is provided to absorb the radiated sound waves.

Following the reviewer's suggestion, we have added more explanations about the simulation setup on **Pages 4-5**: "The simulation is carried out in a cylindrical calculation domain with the designed metasurface positioned at the bottom of the domain, and the spatial lateral extent of the simulation window is selected to be 0.34 m, which is about 7λ beyond the radius of the metasurface. The cylindrical wave radiation condition is applied on the exteriors of the domain to prevent boundary diffraction at the edges, while a perfect matching layer is employed to absorb the radiated sound waves in the far field."

Comment P 1.7 — *Also, an important point is that the bibliography does not give credits to some relevant work of the community:*

First, one of the most prominent researchers of the acoustical vortices community is Pr. Marston. He was the first to transpose the idea of helical wave from optics to acoustics. He was also the first to synthesize experimentally acoustical vortices with four synchronized transducers in ref. [1]. Along with this seminal work, he dedicated a large effort to develop relevant theories in this field, see. e.g. ref. [2*,3*] and I believe that some of his papers are relevant here.*

[1] B. T. Hefner and P. L. Marston, An acoustical helicoidal wave transducer with applications for the alignment of ultrasonic and underwater systems, J. Acoust. Soc. Am. 106, 3313 (1999).*

[2] P.L. Marston. Axial radiation force of a Bessel beam on a sphere and direction reversal of the force, J. Acoust. Soc. Am., 120, 3518 (2006)*

[3] P.L. Marston Scattering of a Bessel beam by a sphere: II helicoidal case and spherical shell example, J. Acoust. Soc. Am. 124:5,2905 (2008).*

Response: We thank the reviewer for raising the outstanding researcher Pr. Marston and his important papers (J. Acoust. Soc. Am. 106, 3313; J. Acoust. Soc. Am., 120, 3518; J. Acoust. Soc. Am. 124:5,2905-10). These references constructed and evaluated a pioneering four-panel transducer capable of producing a beam with OAM and studied the radiation and far-field scattering caused by the acoustic Bessel beam, which are very relevant to our studies. We have added them to the reference list as **Refs. 21-23** on **Page 1**: "There have been substantial efforts to transpose the idea of helical wave from optics to acoustics. Among them, the pioneering design of synthesizing experimentally acoustical vortices with four synchronized transducers was first proposed by Marston et. al. [21], and prominent efforts have been dedicated to develop relevant theories in this field [22, 23] along with this seminal work".

Comment P 1.8 — *Then, the first authors who synthesized an acoustical vortex with an array of transducers were Thomas and Marchiano in ref. [4*] in 2003, long before ref. [20,21] cited in*

the paper. So, this work should be mentioned along with the other two references.

[4*] J.-L. Thomas and R. Marchiano, *Pseudoangular Momentum and Topological Charge Conservation for Nonlinear Acoustical Vortices*, *Phys. Rev. Lett.* 91, 244302 (2003).

Response: We thank the reviewer for pointing out our omission in the reference. In this work (*Phys. Rev. Lett.* 91, 244302), the acoustic vortices were first synthesized by using a network of 55 piezoelectric transducers immersed in water and the conservation of the pseudo angular momentum in a nonlinear regime was experimentally checked. We have added it to the reference list as Ref. 24.

Comment P 1.9 — *In the present paper, the authors propose a method to shape SAW with metascreens in order to generate acoustical vortices. While this idea is undoubtedly original, the idea to use SAWs generated by spiraling holograms to synthesize bulk acoustical vortices was previously proposed in ref. [5*]. In this case, active holograms based on spiraling electrodes were used to shape SAWs instead of the metascreens proposed here. This work should be introduced properly.*

[5*] A. Riaud, M. Baudoin, O. Bou Matar, L. Becera, J.-L. Thomas, *Selective manipulation of microscopic particles with precursor swirling Rayleigh waves*, *Phys. Rev. Appl.*, 7: 024007 (2017)

Response: We thank the reviewer for bringing up this interesting paper (*Phys. Rev. Appl.*, 7: 024007). The paper realized the near-field capture of particles in liquid, which is a breakthrough in the application of surface waves.

Following the reviewer's suggestion, we have added it to the reference list as Ref. 44 in **Page 2**: "Note that synthesizing bulk acoustical vortices in near-field through Rayleigh SAWs generated by spiraling holograms was previously proposed in Ref. 44, in which active holograms based on spiraling interdigitated transducer were used to encode the phase of the field like a hologram and shape SAWs instead of the metasurface proposed here".

Comment P 1.10 — *Finally, concerning particles trapping and manipulation, ref [6*] demonstrated particles manipulation with helical waves with unprecedented selectivity. I suggest to introduce it along with ref. [7-9] of the paper.*

[6*] M. Baudoin, J.-C. Gerbedoen, A. Riaud, O. Bou Matar, N. Smagin, J.-L. Thomas, *Folding a focalized acoustical vortex on a flat holographic transducer: miniaturized selective acoustical tweezer*, *Science Adv.*, 5: eaav1967 (2019)

Response: We thank the reviewer for raising this important reference (*Science Adv.*, 5: eaav1967), which demonstrates the ability of these tweezers to grab and displace micrometric objects in a standard microfluidic environment with unique selectivity. We have added it to the reference list as Ref. 10.

Response to Reviewer 2

Comment P 2.1 — *Sound vortices, which carry quantized topological charges in real space, are very useful in real applications, such as for robust acoustic communication and rotating small particles. This paper provides a novel design route to achieve spiral sound field with long propagation distance. The approach is straightforward and efficient. The theoretical results are well supported by experiments. The paper deserves to be published in Nature Communications. Suggestions and comments are listed below.*

Response: We respectfully thank the reviewer for the valuable and constructive comments, which have helped us a lot in improving our manuscript clarity. We are strongly encouraged by reviewer's report that the paper deserve to be published in Nature Communications. We are delighted to address the items raised by the reviewer.

Comment P 2.2 — *The discussion part should be more concise. Some recurrent sentences should be removed.*

Response: We thank the reviewer for the suggestion. The discussion part has been polished and the recurrent sentences have been removed.

Comment P 2.3 — *I am puzzled about the sentence “Hence the phase singularity still emerges at the core without offset while the hollow part of the central area is extended arising from the variation of the phase”. It is very nice that, in all cases, the phase winding number along a relatively large loop surrounding the weak field region emerges as desired. It seems that each vortex of charge- m is split into m vortices of charge-1, due to the presence of various experimental noise and errors. You can check this effect through a simulation with some fictitious noise, and provide a comment on this issue.*

Response: This reviewer has raised a good point that needs to be clarified to avoid confusion. We are sorry for the unclear statement on the generation of higher-order vortex beams by the planar acoustic antenna. In fact, Figure 5 presents the simulation results of the amplitude and phase distributions of the far-field vortex beams with topological charges $m = 2$ to 4 rather than experimental results. It is found that the phase winding number along a relatively large loop surrounding the weak field region emerges as desired in all cases, but the vortex of charge- m has been split into m vortices of charge-1 rather than one. This phenomenon is consistent with the previous reports that the high-order vortex has a more unstable structure than the first-order vortex, and some slight disturbances could degenerate the unstable high-order vortex into multiple stable first-order vortices [e.g., please refer to Phys. Rev. E 100, 053315 (2019), Appl. Phys. Lett. 116, 163504 (2020)]. In our simulations, the disturbances mainly arise from the weak non-cylindrically symmetric structure design [e.g., see Sci. Adv. 3, e1700552 (2017)] and the interference of the point-like sound source with the radiated acoustic vortex [e.g., see Optics Communications 103, 422-428 (1993) in optics; Phys. Rev. E 71, 066616 (2005), Appl. Phys. Lett. 116, 163504 (2020) in acoustics].

We emphasize this issue in the modified manuscript on **Page 5**: “**indicating that the vortex of charge- m has been split into m vortices of charge-1 rather than one. This phenomenon can be ascribed to effect**

that the high-order vortex has a more unstable structure than the first-order vortex, and some slight disturbances could degenerate the unstable high-order vortex into multiple stable first-order vortices [52, 53]. In our simulations, the disturbances mainly arise from the weak non-cylindrically symmetric structure design and the interference of the point-like sound source with the radiated acoustic vortex [53, 54]”.

Comment P 2.4 — *As a whole, the paper is well organized. However, the language should be polished further. In addition, some terms look strange. For example, “long transmission distance” should be “long propagation distance”. It is better to use “hole depth” and “hole width”, rather than “Air gap depth” and “air gap size”, if the holes are defined to be square.*

Response: The above terms have been revised and we polish the manuscript carefully. We thank the reviewer for the kind suggestion.

Response to Reviewer 3

Comment P 3.1 — *This paper discusses an impedance metasurface formed by cuboid air gap for the generation of acoustic orbital angular momentum (OAM). While the authors claim that they can efficiently excite the OAM field, there is no significant advancement in physics or application. With the existing literature for the efficient generation of OAM both actively using transducer arrays [Hefner and Marston, J. Acoust. Soc. Am. 106, 3313-3316 (1999), Riaud et al, Phys. Rev. Appl. 4, 034004 (2015)] and passively with metasurfaces [Jiang et al, Phys. Rev. Lett. 117, 034301 (2016), Jiang et al, Appl. Phys. Lett. 108, 203501 (2016), Ye et al, AIP Adv. 6, 085007 (2016), Naify et al, Appl. Phys. Lett. 108, 223503 (2016)], I cannot find any improvement compared with these methods. This article does not meet the standard of a high quality journal like Nature Communications. Therefore, I cannot recommend it for publication.*

Response: We thank the reviewer for raising these important references, which provide us an opportunity to clarify the innovation of our proposal and avoid possible confusion. The above mentioned articles are seminal publications enabling efficient generation of OAM. We have read these articles carefully, all of which have been included in the references. And as correctly spotted by the reviewer, existing methods for generating acoustic OAM can be divided into two major categories: active and passive.

The referee mentioned two articles on active scheme: [Hefner and Marston, J. Acoust. Soc. Am. 106, 3313-3316 (1999), Riaud et al, Phys. Rev. Appl. 4, 034004 (2015)] – added to the references. In the two papers, the authors synthesize experimentally acoustical vortices with synchronized transducers, but the aim was to enable acoustic OAM based on arrays of sound sources.

On the other hand, the referee mentioned four articles on passive scheme: [Jiang et al, Phys. Rev. Lett. 117, 034301 (2016), Ye et al, AIP Adv. 6, 085007 (2016), Naify et al, Appl. Phys. Lett. 108, 223503 (2016)] – already in the references, and [Jiang et al, Appl. Phys. Lett. 108, 203501 (2016)] – added to the references. These papers, which all deals with a metasurface, discusses the conversion of

incident plane wavefront into spiral one, here can be regarded as the modulated acoustic transmission systems.

Indeed, the topic of acoustic OAM generation has been widely exploited. However, we would like to emphasize that our strategy offers the possibility for the first time to produce long-distance spiral airborne sound using low-profile planar acoustic antenna. Please note three important differences between the designs in above literatures and our proposal in the manuscript.

(1) Active designs are essentially based on acoustic phased array technique that uses multiple independently regulated acoustic transducers to form the array, producing reconfigurable spatial phase distribution of the desired spiral shape. However, in order to pursue a nearly continuous spiral wavefront for long-distance self-collimated propagation, it is often necessary to depend on a huge number of transducers forming an extremely large-scale acoustic array, and to operate individual units independently through complicated electric system composed of stimulant signal generation and power amplification, ensuring that the phase of the drive signal corresponding to each transducer is precisely modulated and powerful enough. Therefore, the active generation method of acoustic OAM requires large space occupation and complicated circuit, and each transducer unit itself has a certain size, which also brings difficulties to its application in the high-frequency range.

On the contrary, the metasurface we proposed relies on passive technique that overcomes the inherent limitations of complex circuit control associated with the structure, but instead creates a spiral phase distribution required for the vortex field by using a impedance metasurface with a discontinuous impedance gradient, which has the advantages of flat structure and relatively lower cost. Moreover, the fabricated metasurface in the manuscript is composed of 7660 elements, with the deep-subwavelength element period of $d = 6$ mm (that is, about $\lambda/10$, far smaller than the operating wavelength of about 56.7 mm at 6 kHz).

(2) Furthermore, please note that the proposed method is also significantly superior in passive methods. Firstly, our metasurface does not depend on resonances produced by elements with composite internal micro-structure. Secondly, the acoustic diffraction effects require that the dimensions of the structure in the thickness direction should be greater than or equivalent to the acoustic wavelength, while our structure is an unconstrained flat, ultra-thin structure. Although all current planar structures claimed to be thin themselves, we would like to note that the excitation sound sources of incident plane waves and the required propagation distance to avoid near-field effects of practical transducers inevitably take up bulky space in these transmissive configurations, as these designs are based on the modulation of transmitted propagating waves.

On the contrary, what we are focusing on is the modulation and conversion of evanescent spoof surface acoustic waves, which is a guided wave pattern that only propagates along the surface of the rigid periodic structure on the fluid side. We are deliberately investigating the holographic interferogram as a 2D modulated artificial acoustic impedance metasurface and the subsequent complex inter-modal interaction between cylindrical surface waves and the spatially-modulated impedance boundary condition. Such complex interplay cannot be traced in transmission-type metasurfaces mentioned above, indicating that our proposal is **not** a variant of the previous passive structures and they are based on very **different** physics principles. Moreover, our point excitation source can be positioned in the close vicinity of the metasurface, which functions as a coplanar structure as a whole, realizing a **real low profile**.

(3) In addition to the distinct physical mechanism, please note that due to its unique air-compatible conductivity, ultra-thin structure and ability to carry sub-wavelength information to far fields, this metasurface may lead to a wide range of extensive and significant applications. Different from other OAM generators, the air-compatible conductivity allows the sound wave to propagate to a remote distance in the far-field without resorting to the additional waveguides (please note that these waveguides are ordinarily employed in most current metasurface designs for generation and transmission of acoustic vortex, but not applicable or cumbersome in many practical scenarios), which should be promising in non-contact manipulation of particles, acoustic long-distance communication and non-intrusive detection of the energy flux, and the evolution of acoustic waves in the complex fluctuating system.

Indeed, we believe that the entire message of our paper has not been fully appreciated by the referee, which might have been caused by the lack of our presentation. We have gone through the entire letter for improvement. Although we mainly focused on the metasurface for the generation of long-distance spiral beam in the manuscript, the findings of impedance metasurface can enable flexible conversion of spoof surface acoustic waves for realizing versatile functional devices with superior performance. Thus, the proposed strategy provides a feasible approach to design integrated and small-sized system-level antenna as well as being loaded on other functional devices of irregular shape, which can be extended far beyond the scope of original results in the manuscript and utilized in acoustic multi-mode communication as a consequence of the flexible design of impedance surfaces and integration of different modes such as various angular momentums. For example, Figure S1 shows preliminary illustrations of acoustic **dual-frequency OAM-dependent antenna**, which can emit vortices with different charge- m at different frequencies simultaneously. Furthermore, Figure S2 illustrates the acoustic **dual-angle OAM-dependent antenna**, which radiate vortex beams along predesigned horizontal azimuth and pitch angles with different charge- m . These results clearly illustrate the broad application prospects in the field of signal multiplexing. Note that further exemplification of the variety of potential applications is too extensive and beyond the scope of this work.

Hence, as for the presented performance, we believe that our concept display sufficiently new physics with appealing applications. Besides the responses to the reviewer, we would like to emphasize that our impedance metasurface can spark the interest among researchers in this field by building on top of the proposed strategy, which should become a very rich ground for novel discoveries to broad audience of acoustic scientists and engineers. Based on the reviewer's comment, we emphasize this issue in the modified manuscript on **Page 6**: "**Although we mainly focused on . . . detailed discussion can be found in *Supplementary Information Note 5***", and detailed contents have been added in **Supplementary Information Note 5**.

In conclusion, contrary to the referee's opinion, we believe that we have been able to open a new avenue within an established area, displaying a major step forward as compared to the existing literature.

Figure S1: **Demonstration of dual-frequency OAM-dependent antenna.** (a) Schematic diagrams. The spoof SAW are launched by two monopolar sound sources with different frequencies of f_1 and f_2 (yellow pentagram) located adjacent to the metasurface antennas (orange plate), as illustrated by the divergent purple lines. These surface waves carrying zero OAM are scattered into the desired far-field wave beams with different artificial OAM m along z direction. (b) Dispersion curves of unit cells and (c) imaginary part of the surface impedance Z_s with different hole sizes a , while the black line with triangle marks denotes the air line. (d) The structure details and (e) impedance distribution of the designed square metasurface. (f) Simulated phase and amplitude distributions of the helical wavefronts at the observation planes located at 2.00 m away from the metasurface, respectively. The left panels indicate a vortex beam at $f_1 = 5000$ Hz with topological charges $m = 1$, while the right panels show the other vortex beam at $f_2 = 6000$ Hz with topological charges $m = -1$. The white dots refer to the geometric centers of the phase cross sections. See Supplementary Materials *Animation three.mp4* for the dynamic view of dual-frequency OAM-dependent acoustic vortex beam.

Figure S2: **Demonstration of dual-angle OAM-dependent antenna.** (a) Schematic diagrams. The spoof SAW are launched by a monopolar sound source (yellow pentagram) located adjacent to the metasurface antennas (orange plate), as illustrated by the divergent purple lines. These surface waves carrying zero OAM are scattered into the desired far-field wave beams along different horizontal azimuth ϕ_i and pitch angles θ_i . (b) The structure details of the designed circular metasurface. (c) Simulated phase and amplitude distributions of the helical wavefronts at the observation planes located at 2.00 m away from the metasurface, respectively. The left panels indicate a vortex beam along $\theta_1 = 30^\circ$ and $\phi_1 = 0^\circ$ with topological charges $m = 1$, while the right panels show the other vortex beam along $\theta_2 = 30^\circ$ and $\phi_2 = 180^\circ$ with topological charges $m = -1$. The white dots refer to the geometric centers of the phase cross sections. (d) Corresponding far-field radiation pattern. See Supplementary Materials *Animation four.mp4* for the dynamic view of dual-angle OAM-dependent acoustic vortex beam.

REVIEWERS' COMMENTS

Reviewer #1 (Remarks to the Author):

The authors have carefully addressed my concerns. I hence now recommend publication of the paper in Nature Communication.

Reviewer #2 (Remarks to the Author):

I have carefully read all the author's replies and revisions. I think the authors have done an excellent job of revising the manuscript in response to the comments, especially by adding many discussions in the supplementary materials. The satisfactory responses have addressed my concerns and clarified the points of my questions about the manuscript.

In my opinion, this work has opened a new route for constructing artificial acoustic vortices through low-profile metasurface antenna, which can be valuable for applications like far-field acoustic communication and particle manipulation. These features potentially link the current scheme to practical applications with broad readership far beyond current methods, such as enabling frequency and angle multiplexing for acoustic vortex beam. With my prior comment that this study has potential impacts and sufficient innovation, and it is scientifically sound, I strongly support the publication of the manuscript in Nature Communications.

Reviewer #3 (Remarks to the Author):

The authors' arguments are not convincing. Far-field generation of acoustic vortex beams has been well studied in many existing literature including [X. Jiang et al, Phys. Rev. Appl. 14, 034014 (2020)]. Spoof surface acoustic wave is a well-studied field with a lot of existing publications such as [T. Liu et al, Phys. Rev. Appl. 11, 034061 (2019)]. Even though the paper provides some advancement in technical details, it does not match the high standard of Nature Communication, but fits better in more specific technical journals.

Response to Reviewer #1

Comment:

The authors have carefully addressed my concerns. I hence now recommend publication of the paper in Nature Communication.

Response:

We appreciate Reviewer #1 for acknowledging the scientific value of our manuscript. We respectfully thank the reviewer for the valuable and constructive comments, which have helped us in improving our manuscript clarity.

Response to Reviewer #2

Comment:

I have carefully read all the author's replies and revisions. I think the authors have done an excellent job of revising the manuscript in response to the comments, especially by adding many discussions in the supplementary materials. The satisfactory responses have addressed my concerns and clarified the points of my questions about the manuscript.

In my opinion, this work has opened a new route for constructing artificial acoustic vortices through low-profile metasurface antenna, which can be valuable for applications like far-field acoustic communication and particle manipulation. These features potentially link the current scheme to practical applications with broad readership far beyond current methods, such as enabling frequency and angle multiplexing for acoustic vortex beam. With my prior comment that this study has potential impacts and sufficient innovation, and it is scientifically sound, I strongly support the publication of the manuscript in Nature Communications.

Response:

We appreciate Reviewer #2 for pointing out the substantial advances we made in this research field. We respectfully thank the reviewer for the valuable and constructive comments, which have helped us in improving our manuscript clarity.

Response to Reviewer #3

Comment:

The authors' arguments are not convincing. Far-field generation of acoustic vortex beams has been well studied in many existing literature including [X. Jiang et al, Phys. Rev. Appl. 14, 034014 (2020)]. Spoof surface acoustic wave is a well-studied field with a lot of existing publications such as [T. Liu et al, Phys. Rev. Appl. 11, 034061 (2019)]. Even though the paper provides some advancement in technical details, it does not match the high standard of Nature Communication, but fits better in more specific technical journals.

Response:

We thank the reviewer for raising the two latest articles published during the review process, which *separately* enable acoustic vortex beams and spoof surface acoustic wave (SAW). We have read these articles carefully, both of them have been included in the reference list.

The reviewer mentioned the article on acoustic vortex beams: [X. Jiang et al, *Phys. Rev. Appl.* 14, 034014 (2020)]. In this paper, the authors proposed a nondiffractive acoustic vortex beam generated by a reflection-type acoustic metasurface, but the aim was to modulate the OAM symmetry. However, we would like to emphasize that actually their far-field vortex claimed in this reference reaches $z = 7\lambda$ away from the metasurface, which in the strict sense is **far below** the critical distance between the near and far-field (that is, $R^2/\lambda = 36\lambda$ estimated from Fig. 3 of this reference). But noticeably, our metasurface demonstrates a real sense of far-field vortex beam reaching over $z = 35\lambda$ away from the metasurface, which is **far beyond** the critical distance of $R^2/\lambda = 27\lambda$. Moreover, technically their reflection-type metasurface generates **chaotic superimposed** total sound pressure fields consisting of both the incident plane wave along the backward direction and reflected vortex wave along the forward direction. In other words, their vortex beam can only be acquired mathematically through subtracting the incident plane wave from the measured total pressure fields, which inevitably hinder potential application such as particle manipulations. On contrary, owing to the unique scheme of modulating the evanescent spoof SAW, the metasurface we proposed solely radiates a **pure intact** vortex sound field, completely unaffected by any reflections.

On the other hand, the reviewer mentioned another article on spoof SAW: [T. Liu et al, *Phys. Rev. Appl.* 11, 034061 (2019)]. This paper discusses the subwavelength focusing of spoof SAW by a gradient metasurface, which deals with the manipulation of evanescent wave localized at the structure surface. And as spotted by the reviewer, spoof SAW is a well-studied field and this topic of sound control limited in the near field has been widely exploited. However, we would like to emphasize that our strategy offers the possibility for the first time to convert evanescent spoof SAW into propagating far-field spiral vortex with artificial OAM. Not only does this vortex beam arrives at real far-fields in the strict sense, we also demonstrate the capability to out-couple dual-frequency OAMs and dual-angle OAMs capable of signal multiplexing. We foresee that the combination of spoof SAW in the form of evanescent waves with far-field acoustic beams may enable new technological avenues beyond the already far-reaching scientific implications. Hence, as for the presented performance, we believe that our concept displays sufficiently new physics with appealing applications and substantial advancement.

Again, we respectfully thank the reviewer for the valuable and constructive comments, which have helped us in improving our manuscript clarity.